# POST-TRAINING QUANTIZATION FOR VIDEO MATTING

**Tianrui Zhu[1]    Houyuan Chen[4]    Ruihao Gong[2]    Michele Magno[3]**

**Haotong Qin[3*]    Kai Zhang[1*]**

[1]Nanjing University    [2]SenseTime Research    [3]ETH Zürich    [4]HKUST

## ABSTRACT

Video matting is crucial for applications such as film production and virtual reality, yet deploying its computationally intensive models on resource-constrained devices presents challenges. Quantization is a key technique for model compression and acceleration. As an efficient approach, Post-Training Quantization (PTQ) is still in its nascent stages for video matting, facing significant hurdles in maintaining accuracy and temporal coherence. To address these challenges, this paper proposes a novel and general PTQ framework specifically designed for video matting models, marking, to the best of our knowledge, the first systematic attempt in this domain. Our contributions include: (1) A two-stage PTQ strategy that combines block-reconstruction-based optimization for fast, stable initial quantization and local dependency capture, followed by a global calibration of quantization parameters to minimize accuracy loss. (2) A Statistically-Driven Global Affine Calibration (GAC) method that enables the network to compensate for cumulative statistical distortions arising from factors such as neglected BN layer effects, even reducing the error of existing PTQ methods on video matting tasks up to 20%. (3) An Optical Flow Assistance (OFA) component that leverages temporal and semantic priors from frames to guide the PTQ process, enhancing the model's ability to distinguish moving foregrounds in complex scenes and ultimately achieving near full-precision performance even under ultra-low-bit quantization. Comprehensive quantitative and visual results show that our PTQ4VM achieves the state-of-the-art accuracy performance across different bit-widths compared to the existing quantization methods. We highlight that the 4-bit PTQ4VM even achieves performance close to the full-precision counterpart while enjoying $8\times$ FLOP savings.

## 1    INTRODUCTION

The purpose of video matting (Aksoy et al., 2017; Bai & Sapiro, 2007; Chen et al., 2013; Chuang et al., 2001; Feng et al., 2016; Li et al., 2024; Lin et al., 2021; 2022; Sengupta et al., 2020; Sun et al., 2021; Zhang et al., 2021; Zhao et al., 2021; 2022; 2023a;b;c; Yao et al., 2024) is to accurately estimate the alpha matte ($\alpha \in [0, 1]$) of the foreground objects for each frame in a video sequence. The alpha matte defines the foreground opacity at each

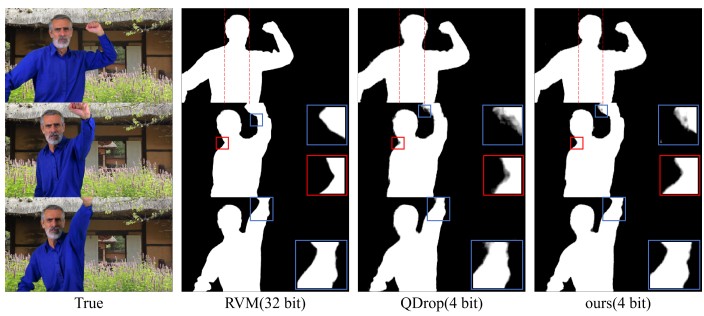

True          RVM(32 bit)          QDrop(4 bit)          ours(4 bit)

Figure 1: Visual comparison of our PTQ4VM against Full-Precision (RVM) and QDrop. Our method demonstrates significant advantages in preserving fine details and temporal coherence.

pixel, governed by the compositing equation $I = \alpha F + (1 - \alpha)B$, where $I$ is the observed pixel, $F$ is the foreground, and $B$ is the background. This challenging computer vision task has broad applications in film production, video conferencing, virtual reality, and more. To enable real-time performance and deployment on resource-constrained platforms for these diverse applications, efficient model representations are crucial. This necessitates advanced model compression techniques to

---

*Corresponding authors

reduce the computational and memory footprint of video matting models, making them suitable for edge computing devices.

Model compression techniques, particularly quantization (Jacob et al., 2018; Nagel et al., 2021; Gholami et al., 2022), are paramount for deploying advanced video matting models on resource-constrained devices by converting high-precision floating-point numbers to low-bit integers, thereby reducing model size and accelerating computation. While Quantization-Aware Training (Qin et al., 2023)(QAT) simulates quantization during training to achieve good performance, it demands extensive labeled data and computational resources, which are often scarce for video matting. In contrast, Post-Training Quantization (PTQ) quantizes pre-trained models directly with minimal calibration data and no retraining, offering significant advantages in deployment efficiency. However, dedicated Post-Training Quantization research for video matting models remains nascent. In this work, we aim to systematically investigate the challenges and opportunities of applying PTQ to video matting tasks.

However, applying PTQ to complex video matting models presents challenges. Firstly, their deep topological structures and the reliance on limited calibration data often lead to unstable convergence during the PTQ calibration process. Secondly, at low bit-widths, quantization errors propagate through the network, resulting in artifacts and increased uncertainty in the output. Furthermore, recurrent structures, crucial for capturing temporal dependencies, are particularly vulnerable to quantization noise, which can destabilize learned temporal dynamics and manifest as flickering or jitter.

To address these challenges, this paper proposes a novel PTQ framework specifically designed for video matting models. To the best of our knowledge, this is the first work to systematically tackle PTQ for this task. Our framework is designed to be general, and its main contributions are as follows:

1. **A Two-Stage PTQ Strategy Combining Block-wise and Global Optimization** We initially quantize the network using block-wise optimization, which achieves fast and stable convergence while capturing critical local dependencies. Subsequently, we perform a global calibration of quantization parameters to minimize accuracy loss while preserving PTQ efficiency.

2. **Statistically-Driven Global Affine Calibration of Quantization Parameters** We observe that neglecting Batch Normalization (BN) layers (Ioffe & Szegedy, 2015) in standard Post-Training Quantization (PTQ) pipelines often leads to significant statistical alterations in the distributions of intermediate layer outputs. We propose a Global Affine Calibration (GAC) method that enables the network to learn a compensation for these cumulative statistical distortions.

3. **Optical Flow Assistance to Guide Post-Training Quantization** To align with the temporal and semantic characteristics of video, we innovatively introduce an Optical Flow (Horn & Schunck, 1981) Assistance (OFA) component. This component utilizes optical flow fields computed from adjacent frames to warp the prediction of the previous frame, serving as a strong temporal and semantic prior for the current frame. Under the guidance of this component, the Post-Training Quantization (PTQ) process enhances the model's ability to distinguish between moving foregrounds and backgrounds in complex scenes.

Our proposed framework (PTQ4VM) not only quantitatively reduces the error of existing PTQ methods on video matting tasks by 10%–20% but also achieves performance remarkably close to the full-precision counterpart, even under challenging 4-bit quantization, while concurrently enjoying substantial $8\times$ FLOP savings, as visually demonstrated in Figure 1 and illustrated in Figure 2.

## 2 RELATED WORK

### 2.1 VIDEO MATTING

**Video Matting** has been significantly advanced by deep learning, surpassing traditional methods (Smith & Blinn, 1996; Chuang et al., 2002). The field leverages diverse architectures, from semantic segmentation models like DeepLabV3 (Chen et al., 2017) adapted for matting, to specialized real-time networks such as BGMv2 (Lin et al., 2021) and MODNet (Ke et al., 2022). These modern approaches are often categorized as assisted or unassisted. Assisted methods, including

OTVM (Seong et al., 2022) and MatAnyone (Yang et al., 2025), require user guidance like trimaps, which limits their automation. In contrast, unassisted methods like RVM (Lin et al., 2022) operate directly on raw video, offering broader applicability. We select RVM as our primary baseline because it represents a widely adopted class of models that balances high accuracy with an efficient encoder-decoder recurrent architecture. To demonstrate our framework's versatility, we also validate it on the Transformer-based MatAnyone, with detailed results in Appendix A.1. This work is motivated by the critical need to compress even efficient models like RVM for deployment on resource-constrained devices.

**Post-Training Quantization (PTQ)** focuses on the accurate determination of the quantization parameters. MSE-based methods are foundational, optimizing $s$ and $z$ by minimizing the Mean Squared Error between the original floating-point tensors and their quantized counterparts using a calibration set. To further enhance PTQ performance, several advanced algorithms have been proposed. AdaRound (Nagel et al., 2020) learns an optimal rounding strategy for weight quantization, adapting weights towards minimizing task loss rather than mere weight reconstruction error, proving particularly effective for very low bit-widths. BRECQ (Block Reconstruction) (Li et al., 2021) improves upon layer-wise quantization by partitioning the network into blocks and optimizing quantization parameters per block to minimize the reconstruction error of its output, thereby better capturing inter-layer dependencies. QDrop (Wei et al., 2022) enhances model robustness to quantization perturbations by simulating quantization noise during calibration, for instance, by randomly dropping quantized versions of activations. While these PTQ techniques demonstrate strong performance on general vision tasks, their optimal combination and adaptation for the unique demands of video matting, such as integrating block-wise optimization with global calibration, specific weight adjustment strategies, and preserving temporal consistency, remain open research areas. Our work addresses these aspects by proposing a tailored PTQ pipeline.

**Optical Flow** estimation computes pixel-level motion between video frames and is widely applied in motion analysis, object tracking, video stabilization, and as input for complex video understanding tasks such as video matting. Traditional methods like Lucas-Kanade (Lucas & Kanade, 1981) rely on local constraints. Deep learning approaches, since FlowNet (Dosovitskiy et al., 2015), learn optical flow end-to-end via CNNs, significantly improving accuracy and robustness. Subsequent methods, such as PWC-Net (Sun et al., 2018), introduced feature pyramids and cost volumes. Among current state-of-the-art algorithms, RAFT (Recurrent All-Pairs Field Transforms) (Teed & Deng, 2020) exhibits outstanding performance. The core of RAFT lies in its iterative optimization mechanism: it constructs a 4D cost volume pyramid of all-pairs correlations and iteratively updates the flow field from an initial estimate using a recurrent unit (e.g., ConvGRU). Key advantages of RAFT include its effectiveness in handling large displacements, strong generalization capabilities, and high accuracy on standard benchmarks. Its iterative nature also allows for a trade-off between accuracy and efficiency. Consequently, we select RAFT to obtain high-precision optical flow priors to assist in the temporal and semantic enhancement of video matting.

## 3 METHODS

### 3.1 PRELIMINARIES

**Weight and Activation Quantization** The fundamental principle of uniform affine quantization maps a full-precision value $v$ (e.g., FP32) to a lower-bit integer $v_q$ (e.g., INT8) using a scale factor $s$ and a zero-point $z$:

$$v_q = \text{clip}(\text{round}(v/s + z), Q_{\min}, Q_{\max}) \tag{1}$$

where $\text{round}(\cdot)$ is a rounding function (e.g., round-to-nearest), and $\text{clip}(\cdot, Q_{\min}, Q_{\max})$ constrains the result to the target integer range (e.g., $[-128, 127]$ for signed INT8). The corresponding dequantization reconstructs an approximation of the original value: $v \approx s(v_q - z)$. The core challenge in PTQ is to find optimal $s$ and $z$ for weights and activations with minimal data and no retraining.

The core optimization objective of weight quantization is to minimize the difference between the original weights $W_{fp}$ and the quantized weights $W_q$. Activation Quantization occurs after the output of activation functions in the network, converting floating-point activations $A_{fp}$ to low-bitwidth integers $A_q$. This process typically uses a small, representative calibration dataset to collect statistical information about activations (such as their range) and thereby determine optimal quantization

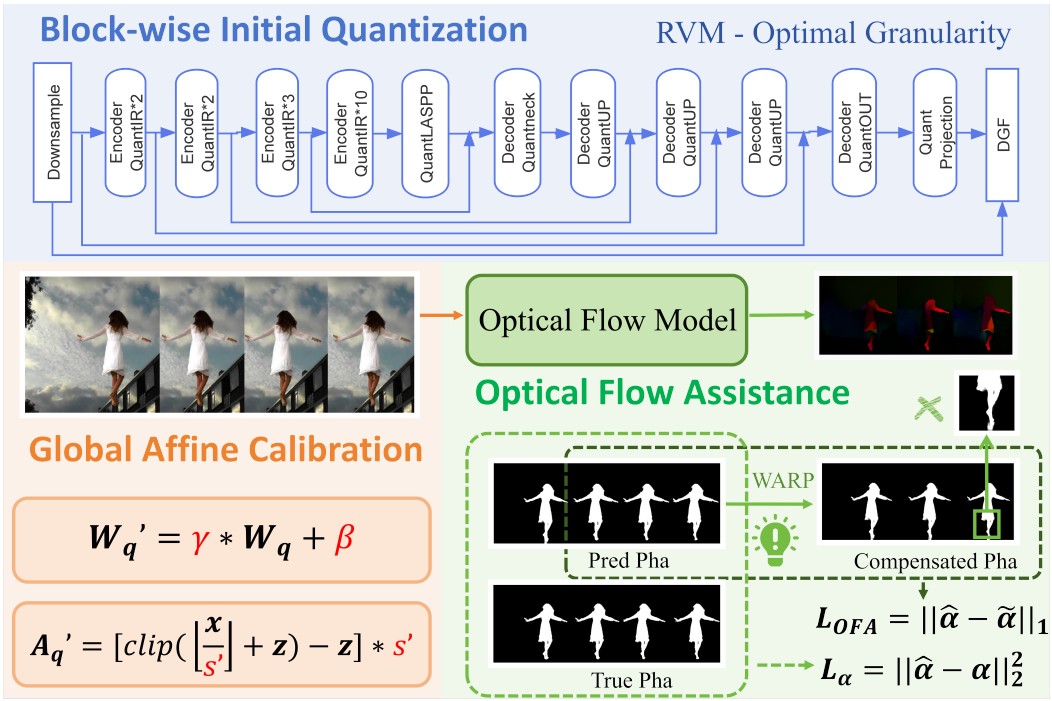

Figure 2: Overview of our PTQ4VM. In Stage 1 (Block-wise Initial Quantization), we optimize functional blocks sequentially to accelerate convergence and enhance stability. In Stage 2, we employ Global Affine Calibration (GAC) to compensate for distributional shifts; meanwhile, Optical Flow Assistance (OFA) guides the model to learn temporal-semantic coherence.

parameters (like the scale factor $s$). The goal is to make the output of the quantized network as close as possible to that of the full-precision network.

**Batch Normalization Folding**  During inference, the operations of a Batch Normalization (BN) layer are linear and can be mathematically equivalent to being fused with the parameters of its preceding convolutional (or fully connected) layer to reduce computation. Let the output of the original convolutional (or fully connected) layer be $Y = WX + B$ (where $W$ are weights, $B$ is bias, and $X$ is input). The subsequent BN layer operation (using fixed parameters at inference) is $Y_{\text{BN}} = \gamma \frac{Y - \mu}{\sqrt{\sigma^2 + \epsilon}} + \beta$, where $\mu$ and $\sigma^2$ are the accumulated mean and variance of the BN layer, $\gamma$ and $\beta$ are learnable scale and shift parameters, and $\epsilon$ is a small constant to prevent division by zero.

Through folding, new equivalent weights $w_f$ and bias $B_f$ can be obtained: $W_f = \frac{\gamma W}{\sqrt{\sigma^2 + \epsilon}}$, $B_f = \frac{\gamma(B - \mu)}{\sqrt{\sigma^2 + \epsilon}} + \beta$ such that the output of the folded layer $Y' = W_f X + B_f$ is mathematically equivalent to $Y_{\text{BN}}$. In full-precision models, this BN folding is lossless. However, during quantization, corrections related to the BN layer are often overlooked. We will discuss this in detail in Section 3.3.

## 3.2  BIQ: BLOCK-WISE INITIAL QUANTIZATION FOR FAST CONVERGENCE & LOCAL DEPENDENCY

**Consideration of Optimization Granularity**  The choice of optimization granularity is a critical factor affecting final quantization performance. When applying PTQ to models with complex architectures, quantization noise can have a significant impact. Some studies (Nagel et al., 2019) indicate that efficient models, particularly those with depth-wise separable convolutions, often exhibit a significant performance drop with PTQ, sometimes even resulting in random-level performance. Our experiments also confirm that attempting direct end-to-end optimization faces challenges such as training instability and convergence difficulties, as detailed in our convergence analysis in Appendix A.2. Concurrently, layer-wise calibration overlooks inter-layer dependencies and can impose high memory requirements, especially in video tasks. We ultimately opted for a block-wise partitioning strategy.

Experiments indicate that block-wise optimization not only excels in computational efficiency and effectively captures crucial local dependencies but also, with appropriate block partitioning, maintains high optimization potential, striking an optimal balance between accuracy and efficiency.

**Block-wise Sequential Optimization and Parameter Learning**    We employ a Dependency-Aware Topological Partitioning strategy. Unlike standard approaches that partition strictly by layer count, we define each computational block ($B_i$) based on functional closure—the minimal topological unit where internal recurrent state updates are self-contained. This ensures temporal integrity is preserved when we quantize blocks sequentially (see Appendix A.2 for details). For the current block $B_i$, the input to its quantized version, $x_{q,in}$, is the output from preceding quantized blocks, while its full-precision counterpart receives $x_{fp,in}$ from preceding full-precision blocks; both originate from the same raw calibration sample. For each block $B_i$, we learn optimal rounding for its full-precision weights $W$ and adaptive scale factors for its input activations. These parameters are determined by iteratively minimizing the Mean Squared Error (MSE) between the block's quantized output $Y_q$ and its full-precision output $Y_{fp}$. This learning process is performed iteratively over the calibration data.

### 3.3    GAC: Global Affine Calibration for Statistical Deviations in PTQ

**Distributional Shift of Intermediate Outputs post-Quantization**    The core issue in PTQ is the significant accuracy degradation post-quantization. We are the first to focus on the Batch Normalization (BN) layer, explaining this phenomenon from a statistical analysis perspective and highlighting the shortcomings of general PTQ frameworks.

Typical Post-Training Quantization (PTQ) frameworks initially fold Batch Normalization (BN) layers into their preceding convolutional or fully-connected layers, yielding effective weights $W_f$. Subsequently, these effective weights $W_f$ undergo weight quantization.

However, we observe that during layer-wise quantization and forward propagation, the errors introduced by weight and activation quantization accumulate. This accumulation leads to a significant shift in the statistical characteristics (e.g., mean, variance, distribution shape) of intermediate layer activations $x$ (i.e., the input to the next layer), causing them to deviate from their counterparts in the full-precision network. When these shifted activations $x$ are processed with the folded weights $W_f$ (which were derived based on original full-precision statistics), $W_f$ is no longer optimal for the actual input distribution it encounters. Consequently, conventional weight quantization strategies aiming to minimize the difference between the original $W_f$ and its quantized version $\hat{W}_f$ become suboptimal, as they fail to account for this input distribution shift.

Critically, such accumulated distributional distortion is further reshaped and altered when passed through non-linear activation functions (e.g., ReLU, Tanh). This poses a significant challenge for subsequent activation quantization steps, as activation quantizers typically employ uniform quantization, relying on simple statistics of the activations, such as observed minimum and maximum values, to determine quantization ranges and scales. When the activation distribution has substantially deviated from its "canonical" or expected form, these statistically driven quantizers struggle to effectively compensate for distortions, potentially leading to considerable accuracy degradation.

**Global Affine Calibration of Dequantized Weights**    Some works (Nagel et al., 2019) have noted the bias in the quantization process and proposed pre-training Cross-Layer Equalization and Absorbing high biases.However, in our experiments, these methods did not yield any performance improvements when applied to the relatively complex models under our investigation. We attribute this primarily to the fact that in complex model architectures, quantization errors propagate layer by layer and are reshaped and amplified by non-linear operations. Consequently, merely adjusting weights quantitatively before quantization struggles to achieve satisfactory results. Therefore, we propose a more general global linear calibration method that directly adjusts the quantized weights.

Our method is as follows: for each convolutional layer $i$ in the network, we introduce two scalar calibration parameters for weights: a scaling factor $\gamma_i$ and a shift factor $\beta_i$. These parameters are applied to the corresponding initially quantized folded weights $W_{f,q,i}$ of that layer:

$$W'_{f,q,i} = \gamma_i W_{f,q,i} + \beta_i \tag{2}$$

A detailed analysis of the learned distributions of these parameters, which empirically validates their role in correcting statistical deviations, is provided in Appendix A.3. Similarly, for activations

$x_i$ input to layer $i$, their representation after applying the quantization function, where $s'_{a,i}$ is the activation scaling factor we optimize and $z'_{a,i}$ is a pre-determined zero-point, can be expressed as:

$$A'_{q,i} = (\text{clip}(\lfloor x_i/s'_{a,i} \rceil + z_{a,i}, Q_{\min,a}, Q_{\max,a}) - z_{a,i}) \cdot s'_{a,i} \qquad (3)$$

where $Q_{\min,a}$ and $Q_{\max,a}$ are the clipping bounds for activation quantization.

The weight calibration parameters $\{\gamma_i\}$, $\{\beta_i\}$, along with the activation scaling factors $\{s'_{a,i}\}$ for all relevant layers, are jointly optimized by minimizing the Mean Squared Error (MSE) between the network's final predicted alpha values ($\hat{\alpha}$) and the ground truth alpha mattes ($\alpha$).

After calibration, these learned parameters $\gamma_i$, $\beta_i$, and $s'_{a,i}$ can be conveniently absorbed into the quantization parameters of the corresponding layer's weights $W_{f,q,i}$ and activations, respectively. Thus, they typically introduce no new parameters or significant additional computational overhead during inference.

This end-to-end optimization process enables the layer-specific $\gamma_i$, $\beta_i$, and $s'_{a,i}$ to collaboratively learn a global compensation mechanism, systematically correcting accumulated errors and distributional shifts introduced by quantization. The method exhibits good universality as it does not rely on complex modeling of specific layers or error types but directly adjusts overall weight and activation scales and biases by optimizing the final task objective. Importantly, our global calibration mechanism can be readily applied on top of various existing PTQ methods, yielding significant performance improvements.

### 3.4 OFA: Optical Flow Assistance for Temporal-Semantic Coherence in PTQ

In video matting tasks, particularly for quantized models, merely predicting $\alpha$ mattes frame-by-frame often fails to capture complex dynamic scenes, leading to temporal flickering or inconsistencies in the output. To further enhance the quality of predictions, we innovatively introduce an optimization method based on optical flow. Optical flow not only provides robust temporal consistency constraints by capturing pixel-level motion trajectories to smooth transitions between consecutive frames, but also assists the model in deeper semantic recognition and motion semantic understanding.

It is noteworthy that although optical flow estimation itself entails a certain computational complexity, which has precluded its direct integration into training scenarios requiring extensive iterations (such as training full-precision models from scratch or Quantization-Aware Training, QAT), Post-Training Quantization (PTQ) typically requires only a very small calibration dataset. This characteristic of low data demand and short training iteration cycles makes the application of optical flow for temporal and semantic enhancement computationally feasible and well-targeted within the PTQ framework.

**Method**    The core idea is to utilize inter-frame motion information to impose temporal constraints on $\alpha$ matte predictions across consecutive frames. Optical flow captures pixel-level motion trajectories between adjacent input frames $I_{t-1}$ and $I_t$. By computing the optical flow field $F_{t-1 \to t}$ from $I_{t-1}$ to $I_t$, the $\alpha$ matte $\hat{\alpha}_{t-1}$ predicted by the model for the previous frame can be effectively warped to the coordinate system of the current frame, yielding a motion-compensated estimate for the current frame's $\alpha$ matte: $\tilde{\alpha}_t = \text{Warp}(\hat{\alpha}_{t-1}, F_{t-1 \to t})$.

This flow-warped matte, $\tilde{\alpha}_t$, serves as a strong temporal prior for the current frame's true $\alpha$ matte. We encourage the model's direct prediction for the current frame, $\hat{\alpha}_t = M_Q(I_t)$ (where $M_Q$ is the quantized model), to align with this motion-compensated prior $\tilde{\alpha}_t$. This alignment is quantified using an L1 loss, which is incorporated as a regularization term into the model's optimization objective, typically for fine-tuning parameters obtained from Phase 1 or during a dedicated PTQ optimization.By pre-computing and storing the optical flow $F$ on the small calibration set, the computation of $\mathcal{L}_{\text{OFA}}$ becomes very concise and rapid. Specifically, since the optical flow is pre-calculated, it causes zero overhead during the actual calibration loop. This lightweight OFA component further enhances the superiority and efficiency of our PTQ framework.

**Procedure and Loss Function**    Given two consecutive frames $I_{t-1}$ and $I_t$ from a video sequence:

1. **Optical Flow Estimation:** Compute the optical flow field $F_{t-1 \to t}$ from $I_{t-1}$ to $I_t$ using the RAFT algorithm.

2. **Previous Frame Alpha Prediction:** Obtain the model's predicted alpha matte for the previous frame, $\hat{\alpha}_{t-1} = M_Q(I_{t-1})$.

3. **Alpha Warping:** Warp $\hat{\alpha}_{t-1}$ using the estimated flow field $F_{t-1 \to t}$ to obtain the motion-compensated alpha matte: $\tilde{\alpha}_t = \text{Warp}(\hat{\alpha}_{t-1}, F_{t-1 \to t})$.

4. **Current Frame Alpha Prediction:** Obtain the model's direct prediction for the current frame, $\hat{\alpha}_t = M_Q(I_t)$.

5. **Optical Flow Assisted Loss:** Calculate the L1 distance between $\hat{\alpha}_t$ and $\tilde{\alpha}_t$ to define the Optical Flow Assisted (OFA) loss: $\mathcal{L}_{\text{OFA}} = \|\hat{\alpha}_t - \tilde{\alpha}_t\|_1$

This loss term $\mathcal{L}_{\text{OFA}}$ is incorporated into the network's overall optimization objective to guide the model (or during a quantization parameter fine-tuning stage) towards generating more temporally coherent and semantically accurate alpha mattes.The effectiveness of this component in reducing temporal errors is experimentally validated in Appendix A.4.

## 4 EXPERIMENTS

We evaluate our method on the VM video matting dataset (Lin et al., 2021) and the D646 image matting dataset (Qiao et al., 2020), with the latter being used to assess generalization as it was unseen during training. For post-training quantization, we use a small calibration set of 256 images sampled from the VM dataset, with further details provided in Appendix A.5. Performance is assessed using standard metrics for alpha matte quality: Sum of Absolute Differences (SAD), Mean Squared Error (MSE), spatial Gradient (Grad), and Connectivity (Conn). For the VM video dataset, we additionally measure temporal coherence using the Deviation of Temporally Smoothed Alpha Differences (DTSSD). Our proposed framework, PTQ4VM, is benchmarked against several state-of-the-art PTQ methods, including a naive MSE-based approach, BRECQ (Li et al., 2021), and QDrop (Wei et al., 2022). For a comprehensive performance reference, we also provide results from several full-precision (FP32) models, including DeepLabV3 (Chen et al., 2017), BGMv2 (Lin et al., 2021), MODNet (Ke et al., 2022), and the original RVM (Lin et al., 2021).

### 4.1 MAIN RESULTS

As shown in Table 1, our PTQ method demonstrates significant advantages across all evaluation metrics on both the VM and D646 datasets. Under the 8-bit quantization setting (W8A8), our method achieves performance comparable to, and in some metrics even superior to, the FP32 full-precision model. In the more challenging 4-bit quantization scenario, where many mainstream PTQ methods exhibit substantial performance degradation or even collapse, our method still maintains satisfactory matting quality and temporal coherence, significantly outperforming other compared methods. For instance, under the W4A4 setting on the VM dataset, our method shows a reduction of approximately 20% in various alpha error metrics compared to the next best method. This robustness at very low bit-widths highlights the superiority of our overall quantization framework in handling complex models and error accumulation. Particularly noteworthy is the performance on the D646 dataset. Since our calibration set is derived entirely from the VM video dataset, D646 represents uncalibrated image matting data for the model. Our method still achieves leading quantization performance on this dataset, which strongly demonstrates the good generalization ability of the proposed method, whose core calibration strategies can be effectively transferred to different data distributions and task characteristics. Overall, our method preserves the accuracy and temporal quality of video matting while substantially compressing model size and reducing computational complexity, providing robust support for the practical application of PTQ techniques in complex video processing tasks.

We also provide visual comparisons. As shown in Figure 3a, our training framework enhances matting accuracy, exhibiting better performance on intricate curve and motion details.Figure 3b demonstrates the improvement in video semantic understanding. Even full-precision models sometimes fail to distinguish similar static background interference, but our model accurately identifies the moving foreground, which also corroborates the guiding role of the OFA component.

To validate versatility beyond CNN-RNNs, we extended our experiments to MODNet (Pure CNN) (Ke et al., 2022) and MatAnyone (Transformer) (Yang et al., 2025), with detailed results provided in

Table 1: Quantitative comparison of our full framework (PTQ4VM) against the FP32 baseline and leading PTQ methods. Our method demonstrates superior performance across various bit-widths on both video (VM) and image (D646) datasets. All metrics are lower the better.

| Dataset | Method | #Bit | #FLOPs | #Param | Alpha ($\alpha$) | | | | | FG |
| | | | (G)↓ | (MB)↓ | MAD↓ | MSE↓ | Grad↓ | Conn↓ | DTSSD↓ | MSE↓ |
|---|---|---|---|---|---|---|---|---|---|---|
| VM 512x288 | DeepLabV3 | W32A32 | 136.06 | 223.66 | 14.47 | 9.67 | 8.55 | 1.69 | 5.18 | – |
| | BGMv2 | W32A32 | 8.46 | 19.4 | 25.19 | 19.63 | 2.28 | 3.26 | 2.74 | – |
| | MODNet | W32A32 | 8.80 | 25.0 | 9.41 | 4.30 | 1.89 | 0.81 | 2.23 | – |
| | RVM | W32A32 | 4.57 | 14.5 | 6.08 | 1.47 | 0.88 | 0.41 | 1.36 | – |
| | RVM-MSE | W8A8 | 1.14 | 3.63 | 6.36 | 1.43 | 1.13 | 0.45 | 1.63 | – |
| | RVM-BRECQ | W8A8 | 1.14 | 3.63 | 6.17 | 1.27 | 1.05 | 0.42 | 1.76 | – |
| | RVM-QDrop | W8A8 | 1.14 | 3.63 | 6.24 | 1.54 | 0.96 | 0.44 | 1.49 | – |
| | Our PTQ RVM | W8A8 | 1.14 | 3.63 | **6.03** | 1.29 | **0.95** | **0.41** | 1.46 | – |
| | RVM-MSE | W4A8 | 0.76 | 2.42 | 168.22 | 158.09 | 14.25 | 24.34 | 4.53 | – |
| | RVM-BRECQ | W4A8 | 0.76 | 2.42 | 28.67 | 19.94 | 7.47 | 3.84 | 3.35 | – |
| | RVM-QDrop | W4A8 | 0.76 | 2.42 | 11.72 | 5.28 | 3.75 | 1.30 | 2.55 | – |
| | Our PTQ RVM | W4A8 | 0.76 | 2.42 | **10.61** | **4.28** | **3.31** | **1.08** | **2.34** | – |
| | RVM-MSE | W4A4 | 0.57 | 1.81 | 189.21 | 184.38 | 15.08 | 27.40 | 3.81 | – |
| | RVM-BRECQ | W4A4 | 0.57 | 1.81 | 168.34 | 161.61 | 15.27 | 24.36 | 5.10 | – |
| | RVM-QDrop | W4A4 | 0.57 | 1.81 | 24.36 | 18.02 | 8.92 | 3.16 | 4.70 | – |
| | Our PTQ RVM | W4A4 | 0.57 | 1.81 | **20.81** | **11.17** | **7.47** | **2.62** | **3.77** | – |
| D646 512x512 | DeepLabV3 | W32A32 | 241.89 | 223.66 | 24.50 | 20.1 | 20.30 | 6.41 | 4.51 | – |
| | BGMv2 | W32A32 | 16.48 | 19.4 | 43.62 | 38.84 | 5.41 | 11.32 | 3.08 | 2.60 |
| | MODNet | W32A32 | 15.64 | 25.0 | 10.62 | 5.71 | 3.35 | 2.45 | 1.57 | 6.31 |
| | RVM | W32A32 | 8.12 | 14.5 | 7.28 | 3.01 | 2.81 | 1.83 | 1.01 | 2.93 |
| | RVM-MSE | W8A8 | 2.03 | 3.63 | 8.03 | 2.56 | 3.22 | 1.97 | 1.10 | 2.77 |
| | RVM-BRECQ | W8A8 | 2.03 | 3.63 | 7.25 | 2.33 | 2.89 | 1.77 | 1.07 | 2.53 |
| | RVM-QDrop | W8A8 | 2.03 | 3.63 | 7.19 | 2.20 | 2.85 | 1.77 | 0.98 | 2.58 |
| | Our PTQ RVM | W8A8 | 2.03 | 3.63 | **7.14** | 2.23 | 2.92 | **1.76** | **0.92** | 2.58 |
| | RVM-MSE | W4A8 | 1.35 | 2.42 | 234.09 | 228.48 | 29.43 | 61.19 | 1.38 | 26.61 |
| | RVM-BRECQ | W4A8 | 1.35 | 2.42 | 60.67 | 50.88 | 18.22 | 15.98 | 1.94 | 16.56 |
| | RVM-QDrop | W4A8 | 1.35 | 2.42 | 19.93 | 11.89 | 10.35 | 5.28 | 1.62 | 4.69 |
| | Our PTQ RVM | W4A8 | 1.35 | 2.42 | **18.77** | **11.14** | **9.94** | **4.97** | 1.61 | 4.97 |
| | RVM-MSE | W4A4 | 1.02 | 1.81 | 234.11 | 228.50 | 29.48 | 61.19 | 1.49 | 11.98 |
| | RVM-BRECQ | W4A4 | 1.02 | 1.81 | 216.46 | 208.53 | 30.24 | 56.64 | 3.77 | 90.92 |
| | RVM-QDrop | W4A4 | 1.02 | 1.81 | 47.91 | 40.15 | 20.85 | 12.60 | 2.36 | 9.13 |
| | Our PTQ RVM | W4A4 | 1.02 | 1.81 | **45.69** | **38.60** | **17.91** | **12.26** | **1.31** | **8.54** |

Table 2: Ablation study of our GAC and OFA components. By incrementally applying them to strong PTQ baselines (BRECQ and QDrop), we demonstrate that each component provides a significant and consistent performance improvement. All metrics are lower the better.

| Dataset | Method | #Bit | #FLOPs | #Param | Alpha ($\alpha$) | | | | | FG |
| | | | (G)↓ | (MB)↓ | MAD↓ | MSE↓ | Grad↓ | Conn↓ | DTSSD↓ | MSE↓ |
|---|---|---|---|---|---|---|---|---|---|---|
| VM 512x288 | BRECQ | W4A8 | 0.76 | 2.42 | 28.67 | 19.94 | 7.47 | 3.84 | 3.35 | - |
| | BRECQ+GAC | W4A8 | 0.76 | 2.42 | **14.91** | **7.21** | **3.37** | **1.73** | **2.50** | - |
| | BRECQ+GAC+OFA | W4A8 | 0.76 | 2.42 | **13.18** | **6.78** | **3.25** | **1.48** | 2.59 | - |
| | QDrop | W4A8 | 0.76 | 2.42 | 11.72 | 5.28 | 3.75 | 1.30 | 2.55 | - |
| | QDrop+GAC | W4A8 | 0.76 | 2.42 | **10.98** | **4.43** | **3.36** | **1.17** | **2.46** | - |
| | QDrop+GAC+OFA | W4A8 | 0.76 | 2.42 | **10.61** | **4.28** | **3.31** | **1.08** | **2.34** | - |
| | BRECQ | W4A4 | 0.57 | 1.81 | 168.34 | 161.61 | 15.27 | 24.36 | 5.10 | - |
| | BRECQ+GAC | W4A4 | 0.57 | 1.81 | **50.75** | **39.84** | **10.44** | **7.11** | 8.01 | - |
| | BRECQ+GAC+OFA | W4A4 | 0.57 | 1.81 | **46.16** | **27.29** | **7.29** | **5.17** | **3.15** | - |
| | QDrop | W4A4 | 0.57 | 1.81 | 24.36 | 18.02 | 8.92 | 3.16 | 4.70 | - |
| | QDrop+GAC | W4A4 | 0.57 | 1.81 | **22.01** | **11.85** | **6.90** | **2.80** | **3.96** | - |
| | QDrop+GAC+OFA | W4A4 | 0.57 | 1.81 | **20.81** | **11.17** | 7.47 | **2.62** | **3.77** | - |

Table 3 of Appendix A.1. Our method maintains high fidelity at 4-bit precision where baselines fail, confirming its robustness across Pure CNN, CNN-RNN, and Transformer architectures.

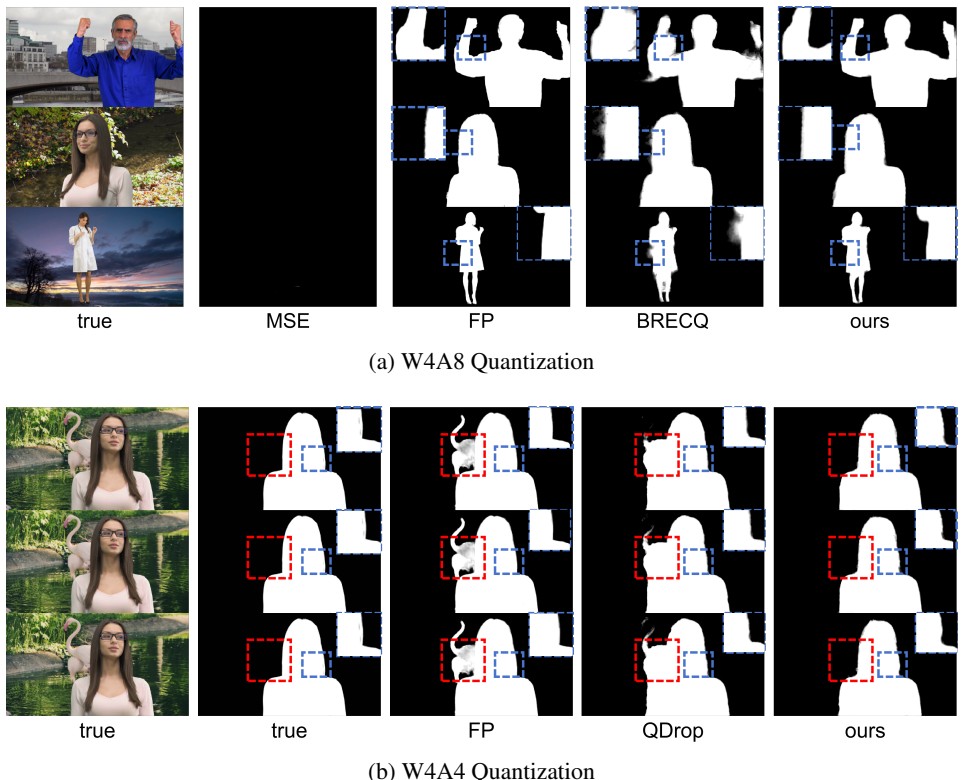

(a) W4A8 Quantization

(b) W4A4 Quantization

Figure 3: Comparison of PTQ4VM with Ground Truth (true), Full-Precision (FP) RVM, MSE, BRECQ, and QDrop under (a) W4A8 and (b) W4A4 quantization. Our method demonstrates superior accuracy and video understanding capabilities.

## 4.2 Ablation Studies

**Effectiveness and Generality Analysis of Global Affine Calibration (GAC)** We apply the GAC module independently to two state-of-the-art PTQ algorithms, BRECQ and QDrop. As shown in Table 2, GAC significantly enhances the performance of both BRECQ and QDrop across various metrics under low bit-width settings, particularly for W4A4. Notably, the performance gain from GAC is particularly significant for BRECQ. After applying GAC, nearly all metrics for BRECQ improve substantially, bringing its performance to a level comparable with QDrop without GAC.

**Effectiveness of the Optical Flow-Assisted (OFA) Component** We investigate the potential benefits of the OFA component for the second-stage calibration of existing PTQ methods. As indicated in Table 2, when the OFA component is integrated into the second-stage calibration process for both BRECQ and QDrop, improvements in accuracy are observed for both methods. This suggests that the temporal priors provided by OFA can effectively guide the optimization.

## 5 Conclusion

This paper presents the first effective Post-Training Quantization (PTQ) framework specifically tailored for the video matting task. We have proposed a general multi-stage quantization strategy that first performs initial quantization via block-wise optimization. Furthermore, we innovatively introduced an Optical Flow-Assisted (OFA) component, which not only significantly enhances the temporal consistency of the quantized model over long video sequences but also improves its video semantic understanding capabilities. Experiments demonstrate that our method can maintain matting quality comparable to full-precision models while substantially reducing model computation and storage requirements, exhibiting superior robustness and generalization even at very low bit-widths.

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

## A APPENDIX

### A.1 GENERALITY VALIDATION

**Rationale and Setup**   To demonstrate that our proposed framework's effectiveness is not limited to a specific architectural style (e.g., the CNN-RNN structure), we conducted extensive validation experiments on two additional models with fundamentally different designs: MODNet (Ke et al., 2022), a pure CNN-based architecture, and MatAnyone (Yang et al., 2025), a state-of-the-art Transformer-based model. This diverse selection allows us to rigorously test the generality of our PTQ approach against distinct architectural paradigms. We applied our method and leading PTQ baselines to both models under 8-bit and 4-bit quantization settings.

**Quantitative Results and Analysis**   The comprehensive quantitative comparison is presented in Table 3. The results clearly show that our framework consistently outperforms other methods across all evaluated metrics and bit-widths on both architectures. At the 8-bit level, our method achieves error rates closest to the full-precision baseline for both MODNet and MatAnyone. The performance gap becomes significantly more pronounced at the challenging 4-bit precision. In this scenario, mainstream methods like MSE and BRECQ experience catastrophic performance degradation or model collapse (particularly evident in the breakdown of MatAnyone and the severe error spikes in MODNet). While QDrop avoids complete failure, it still incurs a substantial accuracy loss. In stark contrast, our framework maintains remarkable stability and accuracy, yielding significantly lower errors and preserving temporal coherence regardless of whether the backbone is CNN or Transformer based.

**Discussion**   The successful application of our framework to both MODNet and MatAnyone strongly validates its generality and robustness. This indicates that the principles behind our method—mitigating local minima through block-wise optimization (BIQ), correcting statistical shifts with global calibration (GAC), and leveraging temporal priors (OFA)—address fundamental challenges in quantization that are not unique to CNN-RNN models but are also prevalent in Transformer-based and pure CNN architectures. This validation supports the conclusion that our proposed framework is a versatile and effective solution for the post-training quantization of a wider range of video matting models.

### A.2 ANALYSIS OF BLOCK-WISE INITIAL QUANTIZATION (BIQ) CONVERGENCE

As discussed in Section 3.2 of the main paper, the choice of optimization granularity is critical to the final performance of Post-Training Quantization (PTQ). This section provides experimental support

Table 3: Quantitative comparison of PTQ methods. We compare the new CNN-based MODNet (blue text indicates added results) with the Transformer-based MatAnyone. Our method consistently outperforms baselines on both architectures.

| Method | Bit | MAD ↓ | MSE ↓ | Grad ↓ | Conn ↓ | DTSSD ↓ | MESSDdt ↓ |
|---|---|---|---|---|---|---|---|
| *MODNet (CNN-based) - New Added* | | | | | | | |
| FP32 Baseline | 32 | 9.41 | 4.30 | 1.89 | 0.81 | 2.23 | 5.50 |
| MSE | 8-8 | 11.25 | 5.12 | 2.45 | 1.23 | 2.98 | 6.45 |
| BRECQ | 8-8 | 10.65 | 4.85 | 2.21 | 1.10 | 2.75 | 6.12 |
| QDrop | 8-8 | 10.10 | 4.62 | 2.08 | 0.95 | 2.50 | 5.85 |
| **Ours** | 8-8 | **9.65** | **4.41** | **1.95** | **0.85** | **2.31** | **5.62** |
| MSE | 4-4 | 152.40 | 85.60 | 12.50 | 18.20 | 8.15 | - * |
| BRECQ | 4-4 | 25.40 | 18.30 | 5.60 | 4.10 | 4.50 | 8.50 |
| QDrop | 4-4 | 15.10 | 7.95 | 3.20 | 1.90 | 3.10 | 6.95 |
| **Ours** | 4-4 | **13.50** | **6.10** | **2.65** | **1.45** | **2.75** | **6.20** |
| *MatAnyone (Transformer-based)* | | | | | | | |
| FP32 Baseline | 32 | 5.15 | 0.93 | 0.67 | 0.26 | 1.18 | 4.78 |
| MSE | 8-8 | 5.87 | 1.23 | 1.01 | 0.48 | 4.72 | 5.28 |
| BRECQ | 8-8 | 5.86 | 1.21 | 0.97 | 0.47 | 5.10 | 5.30 |
| QDrop | 8-8 | 5.62 | 1.16 | 0.80 | 0.41 | 4.87 | 5.16 |
| **Ours** | 8-8 | **5.30** | **1.09** | **0.77** | **0.36** | **4.60** | **5.01** |
| MSE | 4-4 | 171.91 | 170.92 | 14.99 | 28.80 | - * | - * |
| BRECQ | 4-4 | 169.35 | 162.49 | 15.02 | 24.43 | - * | - * |
| QDrop | 4-4 | 20.91 | 17.47 | 7.56 | 3.01 | 4.65 | 6.23 |
| **Ours** | 4-4 | **13.80** | **12.69** | **6.98** | **2.14** | **4.31** | **5.77** |

for this choice by presenting the convergence curves of Alpha error) for block-wise optimization versus naive full-network quantization under different bit-width settings.

**Convergence Comparison under Various Bit-widths**    We compare the convergence process of our proposed Block-wise Initial Quantization (BIQ) method against a naive full-network direct quantization approach (which attempts to optimize quantization parameters for the entire network at once to minimize MSE against the full-precision output, serving as a baseline for comparison) under two different weight-activation bit-width settings: W4A4 and W4A8. The optimization objective for both is to minimize the Mean Square Error (MSE) between the block output (for BIQ) or the final network alpha output (for full-network quantization) and their full-precision counterparts. Figure 4 illustrates the Alpha error, evaluated on the test set, versus the number of iterations for these two settings.

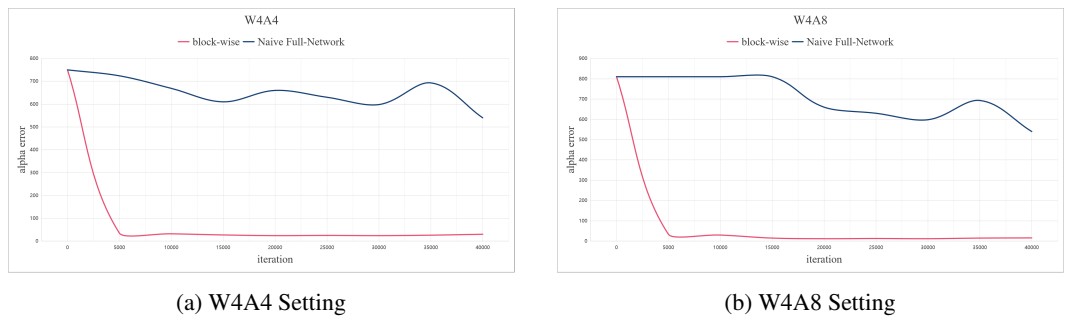

| (a) W4A4 Setting | (b) W4A8 Setting |
|---|---|

Figure 4: Convergence comparison of Alpha error for Block-wise Initial Quantization (BIQ) versus Naive Full-Network Quantization under different settings: (a) W4A4 and (b) W4A8. Evaluations are performed every 5000 iterations, and the curves are smoothed for clarity.

From these convergence curves (Figure 4), we can clearly observe:

- **Effective Convergence of BIQ versus Difficulty of Naive Full-Network Quantization**
  Across the tested bit-widths (W4A4 and W4A8), our Block-wise Initial Quantization (BIQ) method exhibits rapid and effective convergence. The error curve for BIQ drops quickly and stabilizes at a low level within a smaller number of iterations. In contrast, the error curve for the naive full-network quantization method shows little to no significant convergence trend, with its error values remaining persistently high, indicating the difficulty of finding an effective quantization solution by directly optimizing the entire complex network.

- **Superior Final Performance of BIQ** Due to its effective convergence, BIQ achieves a final Alpha MAD value significantly lower than what the naive full-network quantization method can reach (if the latter can be considered to have converged at all). This indicates that by optimizing block by block, we can find a far superior initial solution for the quantization parameters, more effectively capturing local dependencies and avoiding the optimization stagnation or sub-optimal solutions often encountered when attempting to optimize the entire complex network at once.

### A.3 ANALYSIS OF AFFINE CALIBRATION PARAMETER DISTRIBUTIONS IN GAC

To further understand the mechanism by which our Global Affine Calibration (GAC) strategy enhances model performance under various quantization bit-widths (W4A4, W4A8, W8A8), this section provides a detailed analysis of the distribution characteristics of the layer-wise affine transformation parameters learned during the GAC stage: the shift factor $\beta_i$ and the scaling factor $\gamma_i$. Ideally, if the initial quantization stage (e.g., after our first-stage BIQ, or after applying other PTQ methods) had perfectly corrected all statistical deviations, the learned $\beta_i$ would be close to 0 and $\gamma_i$ close to 1. This analysis aims to reveal the extent to which the parameters actually learned by GAC deviate from these ideal values, thereby elucidating the specific compensatory role of GAC for initially quantized models.

**Visualization of Learned Affine Parameters** Figures 5 and 6 respectively illustrate the distribution histograms of the actual $\beta_i$ and $\gamma_i$ parameter values learned for each convolutional layer of the RVM model, and the box plots of their deviations from the ideal values ($\beta = 0, \gamma = 1$), under W4A4, W4A8, and W8A8 quantization settings.

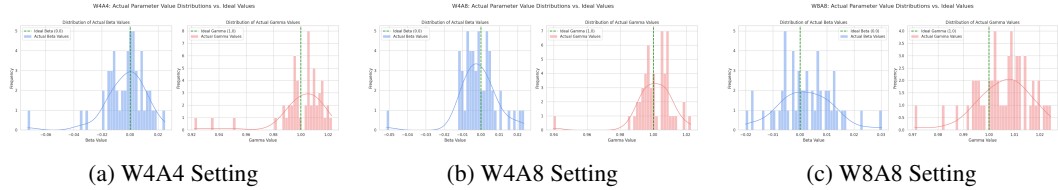

| (a) W4A4 Setting | (b) W4A8 Setting | (c) W8A8 Setting |

Figure 5: Histograms of learned affine calibration parameters $\beta$ and $\gamma$ (each subfigure typically shows distributions for both $\beta$ and $\gamma$) under different quantization settings: (a) W4A4, (b) W4A8, and (c) W8A8. The ideal $\beta = 0$ and $\gamma = 1$ are typically marked for reference within each panel of the subfigures.

**Analysis of Parameter Distributions and Deviations** Figures 5 and 6 collectively reveal the distribution characteristics of the learned affine calibration parameters, $\beta_i$ and $\gamma_i$, and their deviations from ideal values. It is objectively observed from these figures that across all tested bit-widths (W4A4, W4A8, and W8A8), the learned parameters exhibit deviations from their ideal values of $\beta_i = 0$ and $\gamma_i = 1$. Such deviations are particularly pronounced at lower bit-widths, such as W4A4, where the parameter distributions are more dispersed and the absolute range of deviations is larger.

These observed parameter deviations strongly corroborate the presence of significant residual statistical alterations (including both mean shifts and scale changes) in the weight representations after the initial quantization stage, even when advanced strategies like BIQ are employed. The GAC method, by learning non-zero shift factors $\beta_i$ and non-unity scaling factors $\gamma_i$, specifically compensates for these statistical discrepancies. The more pronounced deviations at lower bit-widths further underscore

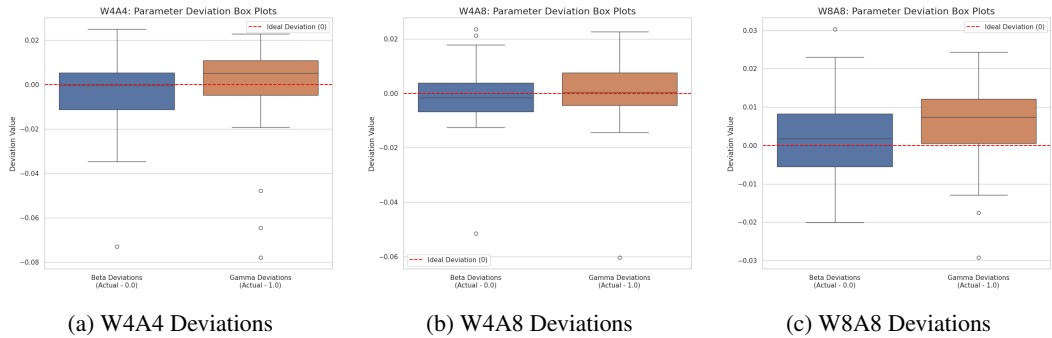

(a) W4A4 Deviations       (b) W4A8 Deviations       (c) W8A8 Deviations

Figure 6: Box plots of deviations for learned affine calibration parameters $\beta$ (from 0) and $\gamma$ (from 1) under different quantization settings: (a) W4A4, (b) W4A8, and (c) W8A8.

the increased importance and efficacy of GAC in calibrating for larger distortions introduced by quantization, thereby explaining its crucial role in model performance recovery.

**Discussion**     The preceding analysis demonstrates that even after employing advanced initial quantization strategies like BIQ, the statistical properties (mean and scale) of the quantized weights in each layer of the network still differ from an ideal state (where no further affine correction would be needed). The Global Affine Calibration (GAC) stage effectively compensates for these residual statistical deviations by learning layer-wise shift factors $\beta_i$ and scaling factors $\gamma_i$. This compensation is particularly crucial for low-bit quantization and is one of the key reasons GAC can significantly enhance the performance of PTQ models. The distributions of these learned parameters, in turn, corroborate the necessity and effectiveness of performing fine-grained statistical calibration within the PTQ pipeline.

## A.4    Effectiveness of the Optical Flow-Assisted (OFA) Component in Calibration

**Experimental Setup**     To further investigate the specific role of the Optical Flow-Assisted (OFA) component during the second-stage calibration process, we conducted a comparative experiment. This experiment, under the W4A4 quantization setting, compares the per-frame average Alpha error when performing joint optimization including the OFA loss term ($\mathcal{L}_{\text{OFA}}$) versus optimization using only the $\mathcal{L}_\alpha$ loss (i.e., without OFA). The experiment was conducted on the test dataset of the VM video dataset, with Alpha errors recorded frame by frame.

**Per-Frame Alpha Error Comparison and Analysis**     Figure 7 illustrates the per-frame average Alpha error curves on the test dataset video sequences for models calibrated with and without the OFA component under the W4A4 quantization setting, with identical parameters used for the BIQ and GAC stages in these experiments to ensure a fair comparison.

As observed in Figure 7, models calibrated with the OFA component (red curve) and without it (green curve) exhibit similar Alpha errors in the initial few frames. However, as the video sequence progresses, the model incorporating the OFA component shows a distinct downward trend in average Alpha error, stabilizing at a consistently lower level. In contrast, the model without OFA maintains a relatively higher error profile throughout the later frames.

This phenomenon clearly demonstrates the effectiveness of the OFA component. Since our OFA loss, $\mathcal{L}_{\text{OFA}}$, is computed and applied to the optimization process starting from the second frame of a video, it leverages temporal prior information provided by optical flow to guide the PTQ calibration. This guidance not only directly encourages the model to learn more temporally coherent representations, thereby reducing prediction errors and instability in subsequent frames, but also indirectly acts as an effective regularizer, aiding the model in achieving higher overall matting accuracy.

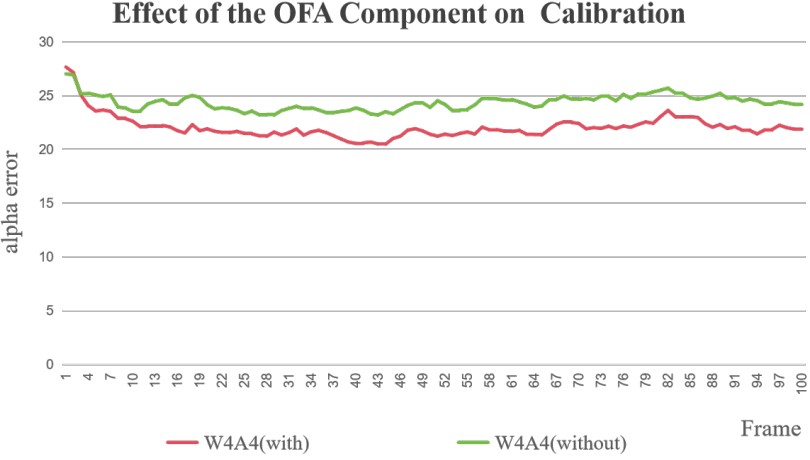

Figure 7: Per-frame average Alpha error comparison for W4A4 quantization with (red curve) and without (green curve) the OFA component on the test dataset.

## A.5 EXPERIMENTAL SETUP DETAILS

**Calibration Set Construction**    As mentioned in the main paper, our calibration set is very small. Specifically, we selected the first 64 video clips from the VM video dataset. For each selected clip, we uniformly sampled frames at indices [0, 2, 4, 6], resulting in a total of $64 \times 4 = 256$ images for calibration.

**Optimization Parameter Settings**    The optimization parameters for our two-stage PTQ framework are set as follows:

- **Stage 1 (BIQ - Block-wise Initial Quantization)** During this stage, for the optimization of each block, we employ the Adam optimizer with a fixed learning rate of $4 \times 10^{-5}$. The number of optimization iterations for each block is set to 20,000.

- **Stage 2 (GAC and OFA)** In this stage, we jointly optimize all learnable calibration parameters, which include the affine transformation parameters $\{\gamma_i, \beta_i\}$ for GAC, the activation scaling factors $\{s'_{a,i}\}$, and implicitly the influence of the OFA loss. The Adam optimizer is used with a unified learning rate of $1 \times 10^{-4}$. The entire calibration process is run for 50 epochs. The weighting factor $\lambda$ for the Optical Flow-Assisted loss term ($\mathcal{L}_{\text{OFA}}$) is set to 0.05.

**Hardware Platform**    All experiments, including model quantization, calibration, and performance evaluation, were conducted on a single NVIDIA RTX 4090 GPU equipped with 24GB of VRAM. It is worth noting that our entire PTQ calibration pipeline has low computational resource requirements, especially in terms of VRAM usage, making it well-suited for typical video matting task scenarios where pre-trained models are efficiently quantized under limited resources.

## A.6 ADDITIONAL TEMPORAL CONSISTENCY EVALUATION

**Evaluation using MESSDdt Metric**    To complement the DTSSD analysis, we employed the MESSDdt metric for a more comprehensive assessment of temporal coherence. This metric evaluates the consistency between model predictions and motion patterns captured by optical flow, offering a distinct perspective on temporal stability.

As illustrated in Table 4, our method maintains consistent advantages across different bit-widths. Standard PTQ approaches exhibit significant performance degradation or complete collapse at lower precision, while the incorporation of our proposed components effectively mitigates these issues.

Table 4: Supplementary quantitative comparison of the MESSDdt metric (↓) on the VM dataset. Using tabularx ensures the background color is continuous.

| Method | FP32 | Quantized (W-A) | | |
| --- | --- | --- | --- | --- |
| | | **4-4** | **4-8** | **8-8** |
| **RVM (Baseline)** | **4.91** | – | – | – |
| MSE | – | -* | -* | 5.31 |
| BRECQ | – | -* | 6.02 | 5.36 |
| BRECQ + GAC | – | **6.80** | **5.98** | 5.40 |
| BRECQ + GAC + OFA | – | **6.34** | **5.81** | **5.19** |
| QDrop | – | 6.23 | 5.87 | 5.31 |
| QDrop + GAC | – | **6.20** | 5.91 | **5.30** |
| **QDrop + GAC + OFA** | – | **6.02** | **5.24** | **4.93** |

The OFA component contributes to measurable improvements in all configurations, bringing the quantized models notably closer to the FP32 baseline performance.

**Robustness to Optical Flow Quality**  We further verified the robustness of our OFA component by testing it with different optical flow estimators, including RAFT (default), GMFlow (transformer-based), and PWC-Net (lightweight). The minimal performance variance across these estimators (Table 5) confirms that our method does not require precise optical flow alignment but rather leverages global motion consistency. This robustness enhances the practical deployability of our approach in resource-constrained environments.

Table 5: Robustness of the OFA component to different optical flow estimators under the W4A8 setting on the VM dataset. The minimal performance variance indicates that our method is not sensitive to the specific choice of the flow model.

| Flow Model | Type | MAD ↓ | MSE ↓ ($\times 10^{-3}$) | DTSSD ↓ |
| --- | --- | --- | --- | --- |
| RAFT (Default) | Recurrent | 10.61 | 4.28 | 2.34 |
| GMFlow | Transformer | 10.55 | 4.21 | 2.32 |
| PWC-Net | CNN (Light) | 10.67 | 4.39 | 2.37 |

## A.7 SUBJECTIVE EVALUATION

To complement the quantitative metrics, we conducted a rigorous Mean Opinion Score (MOS) user study involving 20 participants. The study utilized a blind testing protocol on 20 randomly selected video clips to ensure objectivity. Participants were asked to rate the video quality on a scale of 1 (poor) to 5 (excellent) based on three criteria: **Temporal Stability** (e.g., flickering artifacts), **Boundary Detail** (e.g., hair strands), and **Overall Quality**.

The comparison included three settings: (A) the FP32 Baseline (upper bound), (B) QDrop (W4A4), and (C) Ours (W4A4). As shown in Table 6, the results demonstrate that the QDrop method, while maintaining reasonable boundary details, suffers from noticeable flickering artifacts in the 4-bit setting, leading to lower scores in Temporal Stability. In contrast, our method achieves significantly higher MOS scores, particularly in stability metrics. Participants reported that our method produces visually coherent videos with significantly reduced jitter, perceptually approaching the quality of the FP32 baseline. This subjective preference aligns consistently with our objective MESSDdt and DTSSD improvements reported in the main paper.

Table 6: Subjective User Study Results (MOS). Comparisons are performed on 20 video clips rated by 20 participants. Scale: 1 (Poor) to 5 (Excellent).

| Method | Temporal Stability | Boundary Detail | Overall Quality |
|---|---|---|---|
| FP32 | 4.85 | 4.90 | 4.85 |
| QDrop (W4A4) | 3.10 | 3.45 | 3.25 |
| **Ours (W4A4)** | **4.55** | **4.60** | **4.55** |

### A.8 LLM USAGE STATEMENT

In the preparation of this manuscript, a Large Language Model (LLM) was utilized by the authors.

The role of the LLM was strictly limited to language enhancement and polishing.

Specific tasks included correcting grammatical errors, refining sentence structure for better clarity, and improving the overall readability of the text.

The LLM was not used for generating any core scientific content, which includes but is not limited to: the formulation of research ideas, the development of the methodology, the generation of code, the execution of experiments, and the analysis or interpretation of results.

All intellectual contributions, scientific claims, and conclusions presented in this paper remain entirely the work of the human authors, who bear full responsibility for the final content.

