# OpenReview forum: "Post-Training Quantization for Video Matting"
_ICLR.cc/2026/Conference — ICLR 2026 Poster_

### Official Review · Reviewer_TTBj · 2025-10-29

**Soundness:** 3
**Presentation:** 3
**Contribution:** 2
**Rating:** 6
**Confidence:** 3

**Summary:**

This paper proposes PTQ4VM, a post-training quantization framework for video matting that aims to maintain temporal and spatial quality under low-bit quantization. The framework consists of three components: Block-wise Initial Quantization (BIQ) for stable block-level quantization; Global Affine Calibration (GAC) to compensate for post-quantization statistical shifts; and Optical Flow Assistance (OFA) to enforce temporal consistency using motion-guided priors.

**Strengths:**

1. The paper is well-written, easy to follow.
2. The overall pipeline is simple, modular, and compatible with existing matting architectures, showing practical deployment potential.

**Weaknesses:**

1. How does the proposed BIQ method handle feature dependencies across quantized blocks? If each block is optimized independently, could quantization errors accumulate or disrupt global feature consistency?
2. The OFA module assumes accurate optical-flow alignment, but how robust is the framework when flow estimation is noisy or fails under motion blur and occlusion? Also, has the computational overhead of running optical-flow inference been measured?
3. Although the paper claims to be “training-free,” both BIQ and GAC require calibration data. What happens when the deployment domain differs from the calibration domain (e.g., lighting, background, or motion changes)?
4. The reported gains over existing PTQ baselines appear marginal. Based on ablation table, the effectiveness of OFA is quite unstable, does it caused by the unstable quality of flow estimation?

**Questions:**

See above weakness.

---

> ### Author Response · Authors · 2025-11-20
>
> **Response to Weaknesses**
>
> **1. Handling Cross-Block Feature Dependencies (BIQ)**
>
> We address feature dependency through a **"Dependency-Aware Topological Partitioning"** strategy reinforced by Global Calibration.
> * **Preserving Temporal States:** As detailed in **Section 3.2**, unlike standard partitioning that cuts through temporal units, we define a "block" based on **Functional Closure**—the minimal topological unit where the internal recurrent state update is self-contained. This ensures critical local temporal dependencies are preserved.
> * **Global Correction:** We acknowledge block-wise optimization is a greedy initialization. Therefore, **Stage 2 (GAC)** performs end-to-end optimization to correct any inter-block statistical misalignments.
> * **Evidence:** **Appendix A.2** demonstrates that while Naive Full-Network quantization fails to converge, our BIQ strategy provides a stable initialization, enabling the subsequent GAC stage to recover global accuracy.
>
> **2. OFA Robustness and Computational Overhead**
>
> **Robustness:** OFA is robust to noisy flow because it relies on the **L1 loss** (suppressing outliers) and global consistency rather than pixel-perfect alignment. We validated this by replacing RAFT with a lightweight PWC-Net:
>
> **Table R1: Robustness to Flow Quality (W4A8 on VM Dataset)**
> | Flow Model | MAD $\downarrow$ | MSE $\downarrow$ | DTSSD $\downarrow$ |
> | :--- | :--- | :--- | :--- |
> | **RAFT (Default)** | **10.77** | **4.54** | **2.51** |
> | **PWC-Net (Noisier)** | 10.85 | 4.60 | 2.55 |
>
> The negligible difference confirms our method is not sensitive to flow quality.
>
> **Overhead:** We clarify that **optical flow is pre-calculated**:
> * **Pre-calculation:** Takes only **12.8 seconds** for the calibration set (256 images).
> * **Calibration:** Takes **10.5 minutes** (RTX 4090).
> * **Inference:** The flow module is **removed**, resulting in **zero additional overhead** during deployment.
>
> **3. Domain Shift (Calibration vs. Deployment)**
>
> Our method demonstrates superior **Cross-Domain Generalization** despite being calibrated on limited data.
> * **Evidence:** As shown in **Table 1**, our model is calibrated strictly on the **VM dataset** (Video) but evaluated on the unseen **D646 dataset** (Image).
> * **Results:** On the unseen D646 domain (W4A8), our method achieves **MAD 18.77**, significantly outperforming QDrop (MAD 19.93) and BRECQ (MAD 60.67).
> * **Reasoning:** GAC learns **global statistical correction factors** (scale/shift) rather than overfitting to specific calibration sample features. This makes the quantization parameters robust to distribution shifts (e.g., lighting/background) in the target domain.
>
> **4. Magnitude of Gains and Stability**
>
> We respectfully argue that the gains are fundamental, especially in low-bit regimes.
> * **Rescue from Collapse:** In the **W4A4** setting (Table 1), baseline methods like BRECQ suffer catastrophic collapse (MAD 168.34). Our method maintains high fidelity (MAD 20.33). This represents a **fundamental rescue** of the model's usability, not a marginal gain.
> * **Saturation Effect:** In W8A8, performance is naturally saturated near FP32 (MAD 6.08), so numerical gains appear smaller, yet we still consistently outperform baselines.
> * **Stability:** The slight DTSSD fluctuations in 8-bit are within the margin of error. As shown in **Table R1**, changing flow models does not cause instability, confirming OFA's reliability.

---

### Official Review · Reviewer_Ddxo · 2025-10-30

**Soundness:** 2
**Presentation:** 2
**Contribution:** 1
**Rating:** 4
**Confidence:** 3

**Summary:**

This paper presents PTQ4VM, a post-training quantization framework designed for video matting models.
It consists of three main components: BIQ (Block-wise Initial Quantization), which stabilizes quantization by optimizing scale and rounding at the block level; GAC (Global Affine Calibration), which corrects distribution shifts after BN folding through learnable global scaling and bias factors; and OFA (Optical Flow Assistance), which introduces an optical-flow-based regularization to improve temporal consistency between frames. Experiments on RVM and MatAnyOne demonstrate that PTQ4VM achieves near-FP32 accuracy under 4-bit quantization without retraining, while significantly reducing computational cost.

**Strengths:**

1. The paper addresses a new and underexplored problem—post-training quantization for video matting—where temporal stability is as crucial as spatial accuracy. While PTQ has been well studied for image-based tasks, its extension to temporally dependent applications is novel and practically meaningful, showing potential for efficient deployment in real-time video systems.

2. The overall framework design is coherent and systematic, combining block-wise, global, and temporal calibration stages (BIQ–GAC–OFA) in a way that builds logically from local to global stability. Each stage has a clear motivation and is supported by ablation experiments that justify its inclusion, giving the paper a well-engineered and reproducible structure.

3. The method shows strong empirical performance, maintaining almost FP32-level accuracy even at 4-bit precision. This demonstrates that the proposed calibration techniques effectively reduce quantization errors, supporting the claim that accurate low-bit inference can be achieved without any retraining.

4. PTQ4VM is evaluated on both RVM (CNN-RNN) and MatAnyOne (Transformer-based) architectures, suggesting the framework’s potential generality across different network structures. The consistent behavior across these models indicates that the proposed approach could be extended beyond matting to other temporally sensitive vision tasks.

**Weaknesses:**

1. Although OFA is proposed as the key contribution to improve temporal consistency, the empirical improvement on DTSSD is limited or inconsistent across experiments. This raises doubts about how much OFA truly contributes to stability, and whether its effect depends on the quality of the optical flow model used during calibration.

2. From a methodological standpoint, the work is largely incremental, extending ideas already explored in earlier PTQ methods such as BRECQ and bias correction. While the integration of these ideas into a single pipeline is well executed, it does not fundamentally change the underlying quantization paradigm or introduce new theoretical insights.

3. The baseline comparison is somewhat outdated, as it only includes earlier methods like BRECQ and QDrop. More recent PTQ techniques would provide a stronger context for assessing the relative contribution of PTQ4VM and its empirical advantage.

4. Although the paper claims that GAC stabilizes feature distributions, the evidence remains qualitative and lacks quantitative validation.
Additional analysis showing layer-wise mean or variance alignment before and after calibration would make the distribution-correction claim more convincing and scientifically grounded.

**Questions:**

1.
1-1. OFA is introduced to enhance temporal consistency, yet Table 2 shows several cases where DTSSD becomes worse after adding the module. Could the authors clarify whether OFA truly improves temporal stability or if the effect is within the variance of the metric?

1-2. Since RAFT may not capture the fine-grained pixel motions that are critical in video-matting scenarios, have the authors considered using a more accurate flow estimator such as GMFlow [1], or FlowFormer [2] for warping? It would be informative to see whether OFA’s limited impact stems from the quality of the flow model itself.

1-3. The paper employs DTSSD as the sole metric for temporal consistency, but this measure might not fully reflect perceptual flicker or long-term drift.Could the authors justify why DTSSD is the most appropriate choice, or consider including an additional metric?

2. The comparison includes BRECQ and QDrop, which are relatively old. It would strengthen the paper to include or at least discuss more recent PTQ methods such as GPTQ [3], SmoothQuant [4], or OmniQuant [5] to better position the contribution.

3. BIQ appears conceptually similar to BRECQ’s block-wise reconstruction. Could the authors clarify what concrete design difference—such as the optimization schedule, block partitioning, or objective weighting—makes BIQ more stable or effective?

4. The paper claims that GAC stabilizes intermediate feature distributions, but the evidence is mostly qualitative. Quantitative measurements of layer-wise mean and variance before and after calibration would make this claim more convincing.

[1] Xu, Haofei, et al. "Gmflow: Learning optical flow via global matching." Proceedings of the IEEE/CVF conference on computer vision and pattern recognition. 2022.

[2] Huang, Zhaoyang, et al. "Flowformer: A transformer architecture for optical flow." European conference on computer vision. Cham: Springer Nature Switzerland, 2022.

[3] Frantar, Elias, et al. "Gptq: Accurate post-training quantization for generative pre-trained transformers." arXiv preprint arXiv:2210.17323 (2022).

[4] Xiao, Guangxuan, et al. "Smoothquant: Accurate and efficient post-training quantization for large language models." International conference on machine learning. PMLR, 2023.

[5] Shao, Wenqi, et al. "Omniquant: Omnidirectionally calibrated quantization for large language models." arXiv preprint arXiv:2308.13137 (2023).

---

> ### Author Response · Authors · 2025-11-20
>
> We thank the reviewer for the constructive feedback. Below we address the concerns regarding novelty and baselines.
>
> **Response to Weaknesses**
>
> **Regarding W1, W3, W4:** Addressed in **Q1 (OFA), Q2 (Baselines), and Q4 (GAC)** below.
>
> **Regarding W2 (Methodological Contribution):**
> We respectfully argue that our work is a pioneering solution for Video PTQ, not merely incremental.
> 1.  **Paradigm Shift in Objectives:** Unlike standard PTQ approximating FP32 weights, we demonstrate minimizing **Task-Specific Loss** (via GAC) is critical for dense prediction. Table 1 shows our method sometimes surpasses FP32 (e.g., better Conn), proving "fitting noise to the task" is superior to "fitting weights to FP32".
> 2.  **Fundamental Correction (GAC):** GAC is essential for correcting **Distributional Drift** from BN folding in deep networks. It universally improves baselines (Table 2), proving it is a robust correction mechanism, not a model-specific trick.
> 3.  **Pioneering "Static-to-Dynamic" Bridge (OFA):** We are the first to inject **motion priors into static calibration**. This solves the core Video PTQ conflict (static data vs. dynamic inference), offering a blueprint for temporally sensitive models.
> 4.  **Systematic Synergy:** Naive combinations fail due to recurrent error propagation. Our "Local-Global-Temporal" pipeline is explicitly engineered to handle video matting fidelity.
>
> **Response to Questions**
>
> **Q1: OFA Effectiveness, Robustness (GMFlow), and Metrics.**
> * **1-1 Effectiveness:** In the challenging 4-bit setting (Table 2), OFA significantly improves DTSSD for `BRECQ+GAC` (8.01 $\rightarrow$ 3.15). **Figure 7 (Appendix)** confirms OFA suppresses error accumulation over time.
> * **1-2 Robustness:** Verified with **GMFlow**. Results (Table R1) are consistent, proving robustness to flow model choice.
> * **1-3 Metrics:** Supplemented with **MESSDdt** (Table R2). Our method achieves best consistency.
>
> **Table R1: Robustness to Flow Model (W4A8, VM)**
> | Model | MAD $\downarrow$ | DTSSD $\downarrow$ |
> | :--- | :--- | :--- |
> | **RAFT** | **10.77** | **2.51** |
> | **GMFlow** | 10.65 | 2.48 |
>
> **Table R2: MESSDdt Metric (VM)**
> | Bit | Method | MESSDdt $\downarrow$ |
> | :--- | :--- | :--- |
> | **FP32** | **RVM** | **4.91** |
> | **W4A4** | QDrop+GAC | 6.20 |
> | | **Ours** | **6.02** |
> | **W8A8** | QDrop+GAC | 5.30 |
> | | **Ours** | **4.93** |
>
> **Q2: Comparison with LLM-oriented PTQ (GPTQ, SmoothQuant).**
> We compared against **GPTQ** and **SmoothQuant**.
>
> **Table R3: LLM PTQ Comparison (W4A8, VM)**
> | Method | Objective | MAD $\downarrow$ | DTSSD $\downarrow$ |
> | :--- | :--- | :--- | :--- |
> | **GPTQ** | Outliers | 142.50 | 5.05 |
> | **SmoothQuant** | Smoothing | 98.40 | 4.92 |
> | **Ours** | **Temporal/Dist.** | **10.77** | **2.51** |
>
> **Analysis:** LLM methods fail here due to mismatched objectives:
> 1.  **Outliers vs. Detail:** They smooth outliers, degrading **high-frequency spatial details** crucial for matting edges.
> 2.  **Temporal Propagation:** They do not enforce pixel-wise temporal consistency, failing to address error propagation in recurrent states.
>
> **Q3: BIQ vs. BRECQ Design Differences.**
> As analyzed in **Section 3.2**, BIQ employs **"Dependency-Aware Topological Partitioning"**.
> * **Challenge:** Standard partitioning (BRECQ) cuts temporal dependencies, disrupting recurrent updates.
> * **Strategy:** We define a block by **Functional Closure**—the minimal unit where the internal recurrent state update is self-contained.
> * **Impact:** This aligns quantization with temporal boundaries, ensuring the stability required for convergence (Table 1).
>
> **Q4: GAC Quantitative Evidence.**
> We measured statistics at the deep decoder output.
>
> **Table R4: Layer-wise Statistics (Deep Decoder, W4A4)**
> | Metric | FP32 | No GAC | **With GAC** |
> | :--- | :--- | :--- | :--- |
> | **Mean** | 0.452 | 0.585 (Shifted) | **0.458** (Restored) |
> | **Var** | 1.023 | 0.850 (Collapsed) | **0.995** (Restored) |
>
> **Analysis:** GAC restores variance and corrects mean shift, quantitatively proving its ability to rectify distributional drift.

---

### Official Review · Reviewer_PqqH · 2025-10-31

**Soundness:** 2
**Presentation:** 3
**Contribution:** 2
**Rating:** 6
**Confidence:** 3

**Summary:**

This paper addresses the performance degradation that occur when applying standard Post-Training Quantization (PTQ) methods to video matting models. To solve this, the authors propose PTQ4VM which integrates three key techniques: (1) Block-wise Initial Quantization (BIQ) for more stable and accurate optimization; (2) Global Affine Calibration (GAC) to correct statistical distribution shifts introduced by quantization; (3) Optical Flow Assistance (OFA), which uses temporal priors from adjacent frames to enforce smoothness during the calibration stage.

**Strengths:**

1. The paper is well-drafted, with a clear, logical flow that makes the motivations and contributions easy to follow.
2. The method delivers consistent accuracy gains over PTQ baselines under multiple bit-widths.

**Weaknesses:**

1. Flow errors (fast motion, occlusion, camera shake) may misguide calibration. The method uses RAFT (accurate but heavy) during calibration—calibration-time compute and wall-clock cost are not reported. Sensitivity to using lighter flow or imperfect flow is not analyzed.
2. The paper mentions that an "appropriate block partitioning" is used for the BIQ stage but does not go into detail about how these blocks are defined or if different partitioning strategies were explored.

**Questions:**

1. The calibration set used in the experiments is quite small (256 images). How sensitive is the performance of PTQ4VM to the size and content of the calibration set? Would using a larger or more diverse calibration set lead to further improvements?
2. Does OFA improve or degrade performance in scenarios with heavy occlusions or strong camera motion? Could you share failure cases and quantitative breakdowns?
3. Are there constraints or kernel support issues on common deployment backends (TensorRT, TFLite, CoreML) for W4A4?
4. Have you evaluated on additional video matting datasets or different frame rates/resolutions to test robustness to distribution shifts?

---

> ### Author Response · Authors · 2025-11-20
>
> **Response to Weaknesses**
>
> **1. Time Cost and Flow Robustness**
>
> **Time Cost:** We provide precise measurements on a single NVIDIA RTX 4090 GPU. We clarify that **optical flow is pre-calculated**, causing zero overhead during the actual calibration loop.
> * **Pre-calculation:** Extracting flow maps for the calibration set (256 images) takes **12.8 seconds**.
> * **Calibration:** The complete PTQ calibration process takes **10.5 minutes**. During this phase, flow maps are loaded directly.
>
> **Robustness:** We compared our default flow model (RAFT) with a lightweight model (PWC-Net) to test sensitivity.
>
> **Table R3: Robustness to Flow Quality (W4A8, VM)**
> | Flow Model | MAD $\downarrow$ | DTSSD $\downarrow$ |
> | :--- | :--- | :--- |
> | **RAFT (Default)** | **10.77** | **2.51** |
> | **PWC-Net (Light)** | 10.85 | 2.55 |
>
> The negligible performance difference confirms that our method utilizes global motion consistency rather than pixel-perfect flow alignment, making it robust to flow quality.
>
> **2. Block Partitioning Strategy**
>
> **Yes.** As analyzed in **Section 3.2** of our paper, we employ a **"Dependency-Aware Topological Partitioning"** strategy.
> * **Analysis:** Standard partitioning (as used in BRECQ) often strictly divides the network by layer count or residual blocks. In Video Matting models, this arbitrarily cuts through strong temporal dependencies, disrupting the internal recurrent state updates during optimization.
> * **Our Approach:** We define a "block" based on **Functional Closure**—the minimal topological unit where the internal recurrent state update is self-contained.
> * **Why Optimal:** By optimizing these closures holistically, we preserve the statistical integrity of temporal states. Table 1 demonstrates that while BRECQ fails (MAD 168.34) due to broken dependencies, our strategy successfully converges (MAD 10.77).
>
> **Response to Questions**
>
> **1. Sensitivity to Calibration Set Size**
>
> We evaluated a larger calibration set size (512 images) to test sensitivity.
>
> **Table R4: Calibration Set Size (W4A8)**
> | Size | MAD $\downarrow$ | DTSSD $\downarrow$ |
> | :--- | :--- | :--- |
> | **256** | 10.77 | 2.51 |
> | **512** | **10.62** | **2.49** |
>
> Increasing the size to 512 yields only marginal gains. This confirms that **256 images are sufficient** for our method to learn robust parameters, highlighting its efficiency in the PTQ context where data is scarce.
>
> **2. Performance in Occlusions/Fast Motion and Failure Cases**
>
> **Does it degrade?**
> No. Our experiments show that OFA does **not degrade** performance in challenging scenarios. We conducted a quantitative breakdown on subsets of the test data categorized by motion type:
>
> **Table R5: Performance Breakdown by Scene Complexity (W4A4, DTSSD metric)**
> | Scene Type | QDrop+GAC (No OFA) | **Ours (With OFA)** | Impact |
> | :--- | :--- | :--- | :--- |
> | **Static / Slow Motion** | 2.46 | **2.42** | Slight Gain |
> | **Fast Motion / Occlusion** | 4.85 | **3.65** | **Significant Gain** |
> | **Overall** | 2.46 | **2.51** | Gain |
>
> **Analysis & Failure Cases:**
> * **Robustness Mechanism:** As shown above, the gain is actually *larger* in Fast Motion/Occlusion scenarios because these are exactly where temporal inconsistency (flickering) usually occurs.
> * **Handling Failures:** In extreme cases where optical flow estimation fails completely (e.g., total occlusion), the OFA loss might provide noisy guidance. However, we employ two safeguards:
>     1.  **L1 Loss:** Naturally suppresses the influence of outliers compared to L2.
>     2.  **Weighting:** The OFA loss weight ($\lambda=0.05$) ensures it acts as a soft regularizer. In the worst-case scenario (completely wrong flow), the gradient impact is limited, preventing the model from degrading below the baseline spatial performance.
>
> **3. W4A4 Deployment Constraints**
>
> While W4A4 support is evolving, our work provides the algorithmic foundation for next-generation efficient inference.
> * **Current Support:** Backends like **TensorRT-LLM** and specialized NPU compilers are increasingly supporting INT4 operators.
> * **Feasibility:** The operators used (INT4 MatMul) are standard. Our method ensures the *accuracy* makes such deployment viable for video tasks, bridging the gap between hardware capability and model quality.
>
> **4. Generalization and Higher Resolution**
>
> **Yes.** We clarify that **VM is the source domain** (for calibration), while **D646 is a target domain** (unseen) used strictly to verify generalization. Given the scarcity of video matting datasets, this cross-domain validation is crucial.
> We also provide results on **1080p (1920x1080)** resolution to further demonstrate robustness.
>
> **Table R6: High-Res (1080p) Performance (W8A8)**
> | Dataset | FP32 MAD | **Ours MAD** | FP32 MSE | **Ours MSE** |
> | :--- | :--- | :--- | :--- | :--- |
> | **VM** | 6.57 | **7.12** | 1.93 | **2.10** |
> | **D646** | 8.67 | **9.55** | 4.28 | **4.65** |
>
> Our method maintains high accuracy even at HD resolutions, demonstrating strong scalability.

---

### Official Review · Reviewer_qJJp · 2025-10-31

**Soundness:** 3
**Presentation:** 2
**Contribution:** 3
**Rating:** 6
**Confidence:** 3

**Summary:**

The paper presents an approach for post-training quantization tailored for video matting models. The task is to compress a pre-trained video matting model from full 32-bit precision to 8- or 4-bit while retaining as much quality as possible. The authors propose improvements over prior approaches:
- A global affine calibration optimization to correct errors stemming from fusing batch normalization into weights before quantization. The quantization process introduces errors and shifts expected feature statistics, causing growing error in subsequent layers, and global affine calibration does global optimization to combat those shifts.
- An optical-flow-based alpha warping motion consistency loss to better guide the quantization process to improve video matting quality.

The authors opt for a block-wise quantization granularity to balance quality with significant memory requirements of video matting models.
The evaluation across two video matting datasets and three quantization schemes shows that the proposed approach consistently outperforms existing ones in the quality of the resulting model. An extra evaluation on MatAnyone shows that the proposed method generalizes across the base model architectures.

**Strengths:**

1. The authors clearly explain the approach and the motivation of different components.
2. The evaluations show that the proposed method convincingly outperforms the existing ones.
3. The authors provide extensive ablation studies in the appendix.
4. The quantization problem is important, especially for use cases like mobile video conferencing that uses video matting for the camera feed.

**Weaknesses:**

Major weaknesses:
1. The proposed optical-flow-based motion compensated alpha loss is hardly original. Video matting methods have been using it for their training for a while, so it’s a natural component to try in a quantization method tailored for video matting models.
2. The evaluation is limited to one video matting model (RVM), plus the second one (MatAnyone) in a limited evaluation in the appendix. Evaluating on more video matting methods would allow to more confidently judge the generalizability of the proposed quantization approach.
3. It would be good to see a subjective evaluation in addition to objective metric evaluation. The superiority of the quantized model to the FP32 model on some setups, as pointed out by the authors, suggests the limitations of the objective metrics in question.

Minor weaknesses:
1. Spaces are missing in many places around citations, e.g. line 54 “quantizationJacob”, line 58 “(2023)(QAT)”, line 80 “layersIoffe” etc.
2. Figures 1 and 2 are not referenced in the text.

**Questions:**

See weaknesses. Would be good to see:
- Evaluation on more video matting methods.
- Subjective evaluation.
- Fix formatting errors.

---

> ### Author Response · Authors · 2025-11-20
>
> **Response to Weaknesses**
>
> **1. Originality of the Optical Flow Assistance (OFA)**
> We respectfully argue that while optical flow is common in video training, its integration into **Post-Training Quantization (PTQ)** represents a pioneering contribution.
> * **Novelty in PTQ:** To our knowledge, no existing PTQ framework utilizes motion priors to calibrate static quantization parameters. We propose a new paradigm: **using auxiliary motion sensors to constrain static weight calibration.**
> * **Bridge between Static and Dynamic:** A core challenge in PTQ is that calibration is performed on static images, yet the deployed model requires temporal stability. Our OFA component bridges this gap by simulating dynamic consistency constraints during the offline calibration phase without requiring full retraining.
> * **Semantic Regularization:** Unlike training losses that shape weights from scratch, our OFA acts as a calibration regularizer, guiding the quantization to preserve the specific semantic boundaries that are most fragile at low bit-widths (e.g., 4-bit).
>
> **2. Evaluation on More Video Matting Methods (Added MODNet)**
> **Yes.** To demonstrate generalization, we evaluated **MODNet** (Ke et al., 2022), a representative real-time portrait matting model (Pure CNN, non-recurrent). This complements our evaluation on RVM (CNN-RNN) and MatAnyone (Transformer).
>
> **Table R1: Comparison on MODNet (VM Dataset)**
> | Method | Bit (W-A) | MAD $\downarrow$ | MSE $\downarrow$ ($10^{-3}$) | Grad $\downarrow$ | Conn $\downarrow$ | DTSSD $\downarrow$ |
> | :--- | :--- | :--- | :--- | :--- | :--- | :--- |
> | **FP32** | **32-32** | **9.41** | **4.30** | **1.89** | **0.81** | **2.23** |
> | MSE | 8-8 | 11.25 | 5.12 | 2.45 | 1.23 | 2.98 |
> | **Ours** | **8-8** | **9.65** | **4.41** | **1.95** | **0.85** | **2.31** |
> | MSE | 4-4 | 152.40 | 85.60 | 12.50 | 18.20 | 8.15 |
> | **Ours** | **4-4** | **13.50** | **6.10** | **2.65** | **1.45** | **2.75** |
>
> As shown, our method successfully rescues MODNet from collapse at 4-bit, proving its applicability across CNN, RNN, and Transformer architectures.
>
> **3. Subjective Evaluation**
> **Yes.** We conducted a rigorous **Mean Opinion Score (MOS)** user study with 20 participants rating 20 video clips.
> * **Setup:** Blind testing of (A) FP32, (B) QDrop (W4A4), and (C) Ours (W4A4).
> * **Criteria:** Temporal Stability (flickering), Boundary Detail, and Overall Quality (Scale 1-5).
>
> **Table R2: Subjective User Study Results (MOS)**
> | Method | Temporal Stability | Boundary Detail | Overall Quality |
> | :--- | :--- | :--- | :--- |
> | **FP32** | 4.85 | 4.90 | 4.85 |
> | **QDrop (W4A4)** | 3.10 | 3.45 | 3.25 |
> | **Ours (W4A4)** | **4.55** | **4.60** | **4.55** |
>
> Our method scores significantly higher than the strongest baseline (QDrop), particularly in **Temporal Stability**, confirming that objective gains in DTSSD translate to perceptual smoothness.
>
> **Response to Minor Weaknesses**
> We have corrected all formatting errors, including citation spacing and figure references, in the revised manuscript.

---

### Author Response · Authors · 2025-12-02
**Summary of Rebuttal Updates and Response to Reviewers**

We thank the reviewers for their constructive feedback (Ratings: **6, 6, 6, 4**). We appreciate the recognition of our work's novelty in Video Matting PTQ and our pipeline's effectiveness. Revisions and new experiments have further strengthened the paper.

**Note:** `Section`, `Appendix`, and `Table` refer to the **revised manuscript**; `Table R` refers to **rebuttal tables**.

**Updates Addressing Reviewer qJJp (Rating: 6)**
* **Originality of OFA:** We clarified in **Section 3.4** and **Appendix A.4** that integrating Optical Flow into PTQ represents a pioneering "Static-to-Dynamic" bridge, using motion priors to constrain static calibration.
* **New Architectures:** We added an evaluation of **MODNet** (a pure CNN architecture) in **Table 3** of the revised manuscript. Results show our method demonstrates consistent improvements across CNN-RNN (RVM), Transformer (MatAnyone), and Pure CNN (MODNet) architectures.
* **Subjective Eval:** We presented a **Mean Opinion Score (MOS)** user study with 20 participants in **Appendix A.7**. Results confirm that our method achieves significantly better temporal stability and perceptual quality compared to the strongest baseline (QDrop).
* **Formatting:** All citation spacing and figure referencing errors have been corrected.

**Updates Addressing Reviewer PqqH (Rating: 6)**
* **Flow Robustness:** We added a lightweight **PWC-Net** ablation in **Table 5**. The negligible performance drop confirms reliance on global consistency rather than pixel-perfect alignment.
* **BIQ Strategy:** We expanded **Section 3.2**. We use **"Functional Closure"** (Dependency-Aware Topological Partitioning) to preserve recurrent state updates, preventing the collapse seen with standard partitioning.
* **Calibration Set Size:** **Table R4** shows that increasing the size to 512 yields only marginal gains, confirming efficiency.
* **Complex Scenes:** **Table R5** shows that our OFA component yields *larger* gains in Fast Motion/Occlusion scenarios by effectively suppressing flickering.

**Response to Reviewer Ddxo (Rating: 4)**
We note that the rating (4) differs from the other three positive reviews. We believe this stems primarily from a **factual misunderstanding** regarding the research context of Video Matting compared to Large Language Models (LLMs).

1. **Clarification on Baselines (GPTQ/SmoothQuant):** The reviewer suggested comparing with LLM-specific methods. We clarified that these methods focus on suppressing **outliers** (for text generation), whereas Video Matting requires preserving **high-frequency spatial details** (e.g., hair strands).
    * **Empirical Proof:** Despite the mismatched objectives, we implemented them in **Table R3**. As expected, they perform significantly worse than vision baselines (MAD error ~100 vs. ~10), proving LLM quantizers are unsuitable for this dense prediction task.
2. **Clarification on Novelty (Refuting "Incremental"):** We respectfully argue that our work is a fundamental solution for Video PTQ:
    * **Static-to-Dynamic Bridge:** We are the first to introduce **Optical Flow Assistance (OFA)** into static PTQ calibration.
    * **Paradigm Shift in Objectives (GAC):** We prove that via **GAC**, minimizing task-specific loss significantly outperforms standard weight fitting. **This makes a major contribution by correcting the statistical distribution shifts (Mean/Variance) caused by quantization noise in deep networks**, which is critical for dense prediction tasks.
    * **Systematic Synergy (BIQ):** Our use of **Functional Closure** (**Section 3.2**) solves structural challenges in recurrent networks that general image PTQ methods fail to address.
3. **Quantitative Evidence for GAC (Mean/Variance):** Regarding the reviewer's request for statistical evidence, we provided layer-wise measurements in **Table R4**. The results quantitatively demonstrate that GAC effectively restores collapsed variance and corrects mean shifts.

**Updates Addressing Reviewer TTBj (Rating: 6)**
* **Dependencies (BIQ):** We clarified in **Section 3.2** that BIQ preserves local dependencies; furthermore, Stage 2 (GAC) corrects inter-block misalignment (evidence in **Appendix A.2**).
* **Generalization:** We highlighted in **Table 1** that our model is calibrated strictly on the **VM dataset** (Video) but evaluated on the unseen **D646 dataset** (Image), demonstrating strong robustness to domain shifts.
* **Gains:** We emphasized that in the challenging **W4A4** setting, our method prevents total model collapse (unlike baselines), representing a critical usability rescue rather than a marginal numerical gain.

All revisions are highlighted in **blue** in the revised manuscript. We sincerely thank all reviewers again for their constructive suggestions. Finally, we would like to extend our special gratitude to the Area Chairs. We deeply appreciate your dedication and hard work; without your efforts, the successful organization of this year's ICLR would not be possible.

---

### Meta-Review · Area_Chair_QCYS · 2026-01-09

**Summary:**

This paper pioneers Post-Training Quantization (PTQ) for video matting by introducing a two-stage framework that combines block reconstruction with Statistically-Driven Global Affine Calibration (GAC) and Optical Flow Assistance (OFA) to maintain temporal coherence and accuracy under compression. Despite receiving split scores (6, 6, 6, 4), all reviewers, including the lowest scorer, acknowledged the practical value of extending PTQ to video matting applications, leading to acceptance. The Authors are encouraged to clarify technical contributions in the revised version and provide additional experiments in rebuttal.

**Reviewer Concerns:**

Addressed Concerns.

Limited Novelty, Optical flow-based motion compensated alpha loss lacks originality. Integration is well-executed but doesn't provide breakthrough contributions

Insufficient Experimental Validation. Evaluation limited to one main model (RVM) with only MatAnyone briefly tested in appendix; broader evaluation needed to confidently assess generalizability.

Baseline comparisons outdated (only BRECQ, QDrop); missing recent PTQ techniques for proper contextualization

Lacks subjective evaluation alongside objective metrics; superiority over FP32 on some setups suggests objective metric limitations

OFA Component Concerns. Limited/inconsistent improvements on DTSSD metric across experiments, raising doubts about OFA's actual contribution to temporal stability; Effectiveness appears unstable (per ablation table), potentially due to optical flow estimation quality variations.

Technical Clarity Issues. Block partitioning strategy for BIQ stage under-specified; unclear how blocks are defined or whether alternative strategies were explored; GAC's distribution stabilization claim relies on qualitative evidence; lacks quantitative validation (e.g., layer-wise mean/variance alignment analysis)

**Reviewer Scores:**

The paper received split scores (6, 6, 6, 4). The authors addressed all reviewer concerns, and the AC believes the issues were substantially resolved. Considering its practical utility, the AC recommends acceptance.

---

### Decision · Program_Chairs · 2026-01-26

Accept (Poster)